# Imprinted Genes and Multiple Sclerosis: What Do We Know?

**DOI:** 10.3390/ijms22031346

**Published:** 2021-01-29

**Authors:** Natalia Baulina, Ivan Kiselev, Olga Favorova

**Affiliations:** 1Institute of Translational Medicine, Pirogov Russian National Research Medical University, 117997 Moscow, Russia; kiselev.ivan.1991@gmail.com (I.K.); olga_favorova@mail.ru (O.F.); 2Center for Precision Genome Editing and Genetic Technologies for Biomedicine, Pirogov Russian National Research Medical University, 117997 Moscow, Russia

**Keywords:** multiple sclerosis, parent-of-origin effect, genomic imprinting, *DLK1-DIO3* locus, miRNA

## Abstract

Multiple sclerosis (MS) is a chronic autoimmune neurodegenerative disease of the central nervous system that arises from interplay between non-genetic and genetic risk factors. The epigenetics functions as a link between these factors, affecting gene expression in response to external influence, and therefore should be extensively studied to improve the knowledge of MS molecular mechanisms. Among others, the epigenetic mechanisms underlie the establishment of parent-of-origin effects that appear as phenotypic differences depending on whether the allele was inherited from the mother or father. The most well described manifestation of parent-of-origin effects is genomic imprinting that causes monoallelic gene expression. It becomes more obvious that disturbances in imprinted genes at the least affecting their expression do occur in MS and may be involved in its pathogenesis. In this review we will focus on the potential role of imprinted genes in MS pathogenesis.

## 1. Introduction

Multiple sclerosis (MS) is a chronic autoimmune and neurodegenerative disease of the central nervous system (CNS) in which inflammation, demyelination, and axonal degeneration lead to a fast progression of neurological disability in young adults [1]. MS is a relatively common disease affecting about 1 in 1000 individuals in Europe and North America [2]. In the last few decades, its prevalence has increased substantially due to not only improved diagnostics and survival of patients, but also the rise of MS incidence [3], which contributes to the high social and economic importance of the disease. Demographic studies have shown that MS, like many other autoimmune diseases, is about 2 times more common in women than in men. Moreover, women are diagnosed with MS 1–2 years earlier than men, but men are more likely to develop a more severe MS course.

The clinical course of MS is highly heterogeneous. Most patients have relapsing-remitting MS (RRMS), which is characterized by recurrent acute exacerbations followed by partial or complete recovery, and, with time, they develop secondary progressive MS (SPMS), specified by gradual accumulation of irreversible impairment. Ten to fifteen percent of patients have so-called primary progressive MS (PPMS) with a steady increase of the irreversible neurological dysfunction from onset [4]. To date, RRMS is the most well studied MS course due to its highest prevalence.

MS is a complex disease that arises from interplay between non-genetic and genetic risk factors. Lifestyle and environmental factors, such as Epstein-Barr infection, vitamin D deficiency, smoking, as well as changes in sex hormone profiles are among the best-established non-genetic risk factors of MS [5,6]. The *HLA* locus on chromosome 6 is known as the main marker of genetic susceptibility to the disease; beyond it, more than 200 other loci affecting MS risk are currently identified. At the same time, their cumulative contribution cannot explain more than 48% of MS heritability [7].

Epigenetic changes affecting gene expression in response to external influence represent the link between non-genetic and genetic risk factors, which should be extensively studied to improve the knowledge of MS molecular mechanisms. Epigenetic mechanisms are involved in the establishment of parent-of-origin effects (POEs) that appear as phenotypic differences depending on whether the allele was inherited from the mother or father [8]. The parental transmission of alleles is accomplished by mechanisms other than classical Mendelian segregation of nuclear genes [9].

## 2. Parent-Of-Origin Effects in MS Development

POEs could be determined as effects that arise (1) from the epigenetic regulation of gene expression (such as genomic imprinting, GI—one of the best characterized POE); (2) from the effects of the maternal intrauterine environment on the developing fetus; and (3) from genetic variation in the maternally inherited mitochondrial genome. POEs were until recently almost exclusively discussed in the context of classical diseases of GI, but they are now receiving recognition in a wider range of complex diseases, including MS. Studying POEs in MS is tricky since the establishment of its impact on the disease is influenced by environmental and in utero effects and requires a large population of phenotyped individuals of varying degrees of relatedness, whose genotypes are assigned with parental origin. However it is believed that POEs hold the potential to explain “hidden” heritability to MS.

The increased risk for MS in children of affected mothers was described [10]. A number of studies demonstrated POE for *HLA* locus, where a significant over transmission of *HLA-DRB1**15 from mothers was observed [11,12].

A reciprocal backcross study in rats with experimental autoimmune encephalomyelitis (EAE), the widely-accepted animal model of MS, demonstrated that 37–54% of EAE susceptibility loci depended on parental transmission; these loci overlapped with experimentally confirmed or predicted imprinted genes [13]. Similarly, a study in mice showed that several loci were predisposed for EAE in a parent-of-origin-dependent manner [14].

In this review, we will focus on the potential role of GI as the most well described POE manifestation in MS etiology.

## 3. The Genomic Basis of Imprinting

GI is epigenetically regulated POE in placental mammals that cause monoallelic gene expression. Most of the known imprinted genes are characterized by monoallelic expression in all tissues, but about 28% exhibit monoallelic expression in only one or several tissues, i.e., are imprinted in a tissue-specific manner [15,16,17]. The genes are imprinted depending on the stage of ontogenesis, i.e., imprinted in a stage-specific manner, being biallelically expressed early in development and undergoing only monoallelic expression at later embryonic stages, or vice versa [18,19,20]. For a few imprinted genes a reversal imprinting was demonstrated: The gene is expressed from the maternal allele in some tissues or developmental stages, and from the paternal allele in others [21,22,23].

Many imprinted genes tend to group into extended clusters from hundreds to thousands of bp in length, the so-called imprinted loci, within which there is a coordinated regulation of gene expression [24]. The imprinted loci may include paternal and maternal expressed genes. The structure of the imprinted loci always includes protein coding genes, long noncoding RNA genes, and, commonly, small noncoding RNA genes: MicroRNA (miRNA) and small nucleolar RNA [25]. It is known that genes of non-coding RNA (both long and small) are involved in regulatory processes. Thus, long non-coding RNAs are important regulators of gene expression, organizing nuclear architecture and regulating transcription; they also modulate mRNA stability and translation, and are involved in the process of posttranscriptional modifications in the cytoplasm [26]. MiRNAs, single-stranded short non-coding RNAs, are involved in posttranscriptional regulation of gene expression due to complete or partially complete sequence complementarity between miRNA and target mRNA, which leads to mRNA degradation or inhibition of its translation [27]. Small nucleolar RNAs are mainly involved in posttranscriptional modifications and maturation of rRNA, tRNA, and small nuclear RNAs, as well as in the regulation of alternative splicing [28].

The monoallelic gene expression at the imprinted loci is controlled by independent imprinting control regions (ICRs). ICRs are characterized by the presence of germline differentially methylated regions (DMRs)—CpG-rich sequences, the methylation of which is carried out on one of the parental chromosomes at the stage of gametogenesis [29]. These DMRs direct alternative splicing, regulate the rate of transcription elongation, or select alternative polyadenylation sites, leading to the synthesis of various allele-specific isoforms of transcripts [30,31]. To date, 35 such germline DMRs have been identified in the human genome [32]. In humans most of them are methylated in female gametes, and only three DMRs (in *H19/IGF2*, *MEG3/DLK1* and *ZDBF2/GPR1-AS* imprinted loci) are known to be methylated in male gametes. In addition to these “primary” germline DMRs in the ICRs, imprinted loci can also contain so-called “somatic”, or “secondary” DMRs in which parent-specific methylation is established after fertilization. These “secondary” DMRs are found in the promoters of some imprinted genes or transcription factors’ binding sites [30]. Methylation status of “secondary” DMRs is usually guided by “primary” DMRs.

Long non-coding RNAs [33], insulator proteins [34], and also histone modification [35] take part in the regulation of imprinting together with DNA methylation. Moreover, the products of imprinted genes interact with each other, forming networks, and, thus, participate in a finer tuning of imprinting regulation; it is known that a dysfunction of one imprinted gene can affect other genes expressed from the maternal or paternal alleles [36,37]. The existence of such a network may partially explain the fact that all hereditary GI disorders are characterized by common clinical features, affecting development, growth, behavior, and metabolism [38,39].

By today, disturbances in imprinted genes are found in the pathogenesis of complex diseases, among which cancer is the most studied [40]. Such disturbances may also be involved in the development of several autoimmune and neurodegenerative disorders [41,42,43], including MS [13,44,45]. MS is not a “classic” GI disorder. Nevertheless, it becomes more obvious that disturbances in imprinted genes at the least affecting their expression do occur in MS, as well as in other polygenic diseases, and may be involved in its pathogenesis. Therefore, a promising way of studying MS development may be the search for disturbances in known imprinted genes.

## 4. Imprinted Genes and MS

To analyze the data on the involvement of imprinted genes in the development of MS, we used geneimprint database [https://www.geneimprint.com/], from which 107 human genes with the status “Imprinted” were selected. Of these imprinted genes, 63 are paternally expressed (59%), 34 are maternally expressed (32%); for 10 genes (9%) the imprinted status was either isoform-dependent, random, or unknown. We performed a search of studies (regardless of the year of their publication) in the PubMed database [https://pubmed.ncbi.nlm.nih.gov/], which are indexed by MeSH terms “Multiple sclerosis” and the name of each of these 107 imprinted genes. For further consideration, we selected those genes that were mentioned in publications fulfilling the following criteria: (1) The publication is an original article; (2) the publication contains information on the association of the MeSH-gene with MS and/or with its animal models; (3) biological materials from humans and/or animal model of MS were used to confirm this association. As a result, eight genes with known imprinted status were selected, among which 6 genes, *DLK1*, *DNMT1*, *IGF2*, *MEG3*, *PLAGL1*, and *ZFAT* are paternally expressed (75%), and 2 genes, *RB1* and *WT1,* are maternally expressed (Table 1). Here we will consider the association with MS of all these genes. Based on the genomic organization they could be divided into those that are components of imprinted loci or single-imprinted. Of all these genes, only 3 are located in imprinted loci: *DLK1* and *MEG3* are clustered in *DLK1-DIO3* locus, and *IGF2* in *IGF2-H19* locus. Due to the common mechanisms of regulation in an imprinted locus, we will also highlight the currently known data on the involvement of other components of *DLK1-DIO3* and *IGF2-H19* loci in MS, since it may promote the interest for understanding the role of GI in MS.

### 4.1. The Association of DLK1-DIO3 Locus with MS

Two of the imprinted genes associated with MS, namely, *DLK1* and *MEG3*, are located in the *DLK1-DIO3* locus, for which POE in EAE mice model was described [13]. This locus is mapped in humans to chromosome 14 (14q32.2) and known to play an important role in prenatal development, placenta formation, skeletal and muscle development, postnatal metabolism, and brain functioning [46]. Figure 1 represents the schematic structure of *DLK1-DIO3* locus. It contains three protein-coding genes preferentially expressed from the paternal chromosome—*DLK1*, *DIO3*, and *RTL1*. The *DLK1* gene (Delta Like Non-Canonical Notch Ligand 1) is located at the 5′ end of the locus and encodes a protein of the epidermal growth factor-like repeat-containing family, that is able to bind NOTCH1 and suppresses its activation and signaling [47]. The *DIO3* gene flanks the 3′ end of the locus and encodes type III iodothyronine deiodinase that is involved in the control of thyroid hormone homeostasis by converting prohormone T4 and active hormone T3 into metabolites with the low affinity for the thyroid hormones nuclear receptors—3,3′,5′-triiodothyronine and 3,3′-diiodothyronine, respectively [48]. The allelic expression pattern of *DIO3* varies across tissues during human ontogenesis and, moreover, is transcript-specific: Its biallelic expression was shown in the placenta and few other tissues [49,50], paternal *DIO3* expression is established in human newborn tissues [51], similar to that observed in the mouse fetus [52,53], and its maternal expression occurred in adult skin biopsy, which expresses a larger mRNA transcript [51]. Importantly, the degree of preferential paternal *Dio3* expression varies significantly across newborn brain regions in the mouse, being strongest in the hypothalamus and moderate in the cerebral cortex, hippocampus, and striatum [51]. The *RTL1* gene (Retrotransposon Gag Like 1) encodes the Retrotransposon-Like Protein 1, which plays an important role in the capillaries of endothelial cells, participating in the establishment of the feto-placental barrier and the development of the placenta.

As seen in Figure 1, in addition to protein-coding genes, the locus *DLK1-DIO3* contains a number of long non-coding RNA genes (*MEG3*, *MEG8*, *MEG9*, and *RTL1AS*), several large clusters of miRNA genes (10 miRNA genes in 14q32.2 and 44 miRNA genes in 14q32.31 clusters) and small nucleolar RNA genes (*SNORD112*—one gene, *SNORD113*—nine paralogous genes, and *SNORD114*—31 paralogous genes), expressed, on the contrary, from the maternal chromosome. *MEG3*, *MEG8*, and *MEG9* genes (Maternally Expressed Gene 3, 8, and 9) encode non-protein-coding RNAs 3, 8, and 9, respectively. *RTL1AS* gene encodes antiRTL1 long non-coding RNA—fully complementary antisense transcript of *RTL1*, which acts as its transcriptional repressor. MEG3 is assumed as tumor suppressor, regulating gene expression via chromatin modification, transcription, and posttranscriptional procession [54]. The essential role of MEG8 in the TGF-β–induced epithelial-mesenchymal transition program was shown in multiple types of cancers [55]. MEG9 was established as a lncRNA with protective role in tumor angiogenesis, which action is induced by DNA damage [56].

Summarizing, protein-coding and non-protein-coding imprinted genes from *DLK1-DIO3* locus are characterized by different POEs: The first ones are preferentially transcribed from paternal, while the second—from maternal alleles. It is deemed that monoallelic expression of these genes, depending on the parent-of-origin, is controlled by differential methylation in several regions, such as the “primary” intergenic IG-DMR [32] and the “secondary” MEG3-DMR [57] and MEG8-DMR [58]. IG-DMR and MEG3-DMR are methylated on the paternal chromosome and are not methylated on the maternal, while MEG8-DMR, on the contrary, is methylated on the maternal chromosome. Concurrently, a recent study demonstrated that imprinted gene expression at the *Dlk1-Dio3* cluster in mice is also regulated by an intricate transcriptional regulatory landscape, involving multiple regulatory sequences that are interpreted in a tissue-specific fashion [59].

Data on the involvement of the *DLK1-DIO3* locus in the development of MS is limited, however components of this locus have already drawn special attention as possible contributors to MS development [45].

*DLK1* gene: There is every reason to believe that this gene is involved in the development of MS. Modern outlooks about MS pathogenesis suggest the active participation of immune cells (primarily T and B cells, as well as natural killers and monocytes), the activation of which at the periphery leads to the development of autoimmune inflammation in the central nervous system; this causes damage to the myelin sheath (demyelination), loss of axons and oligodendrocytes, neurodegeneration, and leads to irreversible neurological dysfunction [60,61]. Several studies have shown the participation of the *DLK1* gene in the development and functioning of the immune system, including the B-cell response [62,63] and pro-inflammatory activation of macrophages [64], as well as in the regulation of the expression of several immune-related genes, including genes of some pro-inflammatory cytokines, and gene of transcription factor gene NfkB [65]. In addition, *DLK1* can act as an atypical Notch ligand that can inhibit the Notch signaling pathway [47,66].

An association of the *Dlk1* with the development of EAE was established in rats. Paternally transmitted risk allele accounted for lower expression of *Dlk1* imprinted gene from this locus in spleen, in both the EAE susceptible and the EAE resistant rats, and in their reciprocal hybrids [13]. Taking into account data from transgenic mice overexpressing *Dlk1*, it was concluded that reduced *Dlk1* expression drives more severe disease and modulates adaptive immune reactions in EAE [13].

Long non-coding RNA genes: As mentioned above, a number of long non-coding RNA genes (*MEG3*, *MEG8*, *MEG9*, and *RTL1AS*) are present at the *DLK1-DIO3* locus (see Figure 1). Of these, *MEG3* is significantly downregulated in the whole blood of MS patients when compared to healthy donors [67]. The possible role of *MEG3* in MS may arise from the ability of this lncRNA to modulate inflammatory response via sponging of miR-138, which regulates IL-1β level [68]; IL-1β is known to participate in EAE and MS [69]. *MEG3* also seems to modulate CD4+ T cell proliferation and IFN-γ and TNF-α levels [70], levels of genes which products are involved in TGF-β signaling pathway [71], and regulates the response of endothelium to the DNA damage [72].

When analyzing the expression of 90 long non-coding RNAs in peripheral blood mononuclear cells (PBMC), decreased expression of *MEG9* was observed in RRMS patients compared to the control group [73]. Along with *MEG3*, maternally expressed non-protein-coding RNA 9 encoded by *MEG9* gene, plays a protective role in tumor angiogenesis in response to DNA damage [56].

MiRNA genes: The involvement of miRNAs from *DLK1-DIO3* locus in MS development is better studied. Several studies reported on changes in expression of some individual miRNAs encoded at this locus in different biological material of patients with various MS courses (Table 2). Almost all studies from Table 2 except one based on RNA-seq data [44] were not focused on investigation of the entire miRNA cluster from *DLK1-DIO3* locus, but on individual miRNAs. Many studies investigated serum as biological material, which is not surprising since there is a big potential in using miRNAs as biomarkers of MS progression or treatment response. Several miRNAs are dysregulated in different biological material, which most likely indicate a variety of their functions. For some miRNAs a biological role was discovered. In more detail, the upregulation of miR-432-5p in MS grey matter lesions was identified when compared to white matter lesions, and, along with four other miRNAs, miR-432-5p modulates neuronal structures in MS by targeting synaptotagmin-7 [74]. The lower miR-485 level in CD4+ T cells of RRMS patients compared to healthy individuals was shown to be correlated with the mRNA and serum level of survivin, which is involved in regulation of apoptosis and cell survival [75]. Our recent study suggests that the vast majority of miRNAs encoded at the *DLK1-DIO3* locus is dysregulated in MS: We detected the increased expression of 43 miRNAs from the locus in PBMC of male RRMS patients compared to healthy men [44]. Network-based enrichment analysis showed that signaling pathways activated by receptor tyrosine kinases were significantly enriched with proteins that are encoded by targets of these differentially expressed miRNAs. These pathways based on Reactome hierarchy are involved in the signaling via stem cell factor, fibroblast growth factor receptor, platelet-derived growth factor, receptors tyrosine-protein kinase ERBB-2 and ERBB-4, neurotrophic receptor tyrosine kinase 1, epidermal growth factor receptor, and insulin like growth factor 1 receptor [44]. The levels of miR-127-3p, miR-370-3p, miR-409-3p, miR-432-5p, as well as miR-376c-3p significantly differed in the serum of PPMS and SPMS patients when compared to healthy controls [76,77], while the serum levels of miR-433-3p, miR-485-3p, and miR-432-5p were shown to differ in RRMS patients compared with PPMS and SPMS patients [77]. Expression of miR-494 was significantly lower in T cells of RRMS patients when compared to the control group [78]. MiR-337-3p negatively correlated with the EDSS in RRMS and SPMS patients [79]. In the context of animal models, the expression of miR-127 and miR-136 from the *DLK1-DIO3* locus was increased in rats predisposed to EAE compared to EAE-resistant rats [80], which completely coincides with the data obtained for MS [44]; increased expression of other miRNAs from this locus, such as miR-434, miR-541, and miR-369, was also detected in rats predisposed to EAE [80]. Overexpression of miR-134-3p in EAE rats promoted CD34 + cell proliferation via inhibition of serine protease 57 [81]. In reactive astrocytes, miR-409-3p affected SOCS3/STAT3 pathway and thus induced the production of inflammatory cytokines, enhancing astrocyte-directed chemotaxis of CD4+ T cells, and leading to EAE exacerbation in mice [82].

The fundamental point is the fact that the observed increased expression of miRNA genes from the *DLK1-DIO3* locus in [44] was characterized by sexual dimorphism and was observed only in men. Although the mechanisms underlying the differences in miRNA expression according to sex in RRMS remain unclear, recent studies indicate that the X chromosome and sex hormones may play an important role in its modulation [87,88,89]. The data about the sex-specific miRNA expression from the *DLK1-DIO3* locus is very limited. It was shown that the estrogen-related ERRγ receptor may regulate transcription of the *MIR433* and *MIR127* genes from the locus [90]. Cis-miR-eQTL SNP rs4905998 was shown to be associated with allele-specific expression of 16 miRNAs from the *DLK1-DIO3* locus, while its proxy SNP rs6575793 is associated with the age of menarche [91].

It is interesting to note that changes in expression of miRNA from *DLK1-DIO3* locus were also mentioned in studies devoted to MS treatment. A decrease in the expression level of miR-411* was observed in the peripheral blood of treatment-naïve RRMS patients when compared with RRMS patients upon natalizumab treatment [84]. The assessment of miRNA profiling in peripheral blood of MS patients treated with fingolimod identified increased level of miR-381-3p in fingolimod responders compared to healthy controls, while miR-655-3p level was lower in both fingolimod responders and non-responders compared to controls [83].

Overall, miRNAs from the *DLK1-DIO3* locus are extensively involved in the development of MS at different levels: They are associated with clinical diversity, activity of pathological processes, and treatment response.

DNA methylation pattern in *DLK1-DIO3* locus: Due to the fact that for the genes located in the *DLK1-DIO3* locus, the association with cancer development is shown, the methylation of DMRs of this locus is usually studied in cancer patients [92,93,94]. In patients with autoimmune pathology, a targeted analysis of the methylation of the *DLK1-DIO3* locus has not yet been performed.

In MS patients, DNA methylation is usually studied using high density DNA methylation arrays and high throughput sequencing. The use of these methods allows the detection of DMRs throughout the genome, however, it has a number of limitations associated with their relatively low resolution. All such studies were performed on mixed groups of male and female patients. In most of them, differential methylation of the *DLK1-DIO3* locus was not detected in CD4+ T-lymphocytes [95,96,97,98], CD8+ T-lymphocytes [99,100], CD19+ B-lymphocytes [101], and CD14+ monocytes [102]. However, in a recent large study involving all four of the mentioned leukocyte populations [103], when analyzing DNA from CD19+ B lymphocytes of RRMS patients, significant differences in the levels of DMR methylation in *MEG3*, *MEG8*, and *RTL1* genes were found when compared with the control group. The authors of the study focused on the search for differential methylation markers that are universal for different leukocyte populations, and therefore excluded these DMRs from further analysis. However, the obtained results indicate the need for further study of the methylation of the *DLK1-DIO3* locus in order to establish the role of this epigenetic mechanism in the regulation of gene expression from this locus in MS.

It is noteworthy that imprinted genes tend to contain sex-specific CpG islands than unimprinted ones. A meta-analysis revealed significant associations of sex-specific methylation of CpG islands in the *MEG3* gene [104]. Based on these data, it can be assumed that the observed sex-specific nature of the expression of miRNAs from the *DLK1-DIO3* locus in RRMS can also be explained by sex differences in the methylation of imprinted regions in the disease.

### 4.2. The Association of Imprinted Genes from IGF2-H19 Locus with MS

The *IGF2* gene is mapped to chromosome 11 (locus 11p15.5) and encodes insulin like growth factor 2. It is located in *IGF2-H19* imprinted locus (Figure 2) which also harbors imprinted gene *H19* with shared enhancers, and *cis*-acting regulatory elements, such as the ICR. The *IGF2* gene is paternally imprinted, whereas *H19* is maternally imprinted. The activation of *IGF2* expression occurs when “primary” H19-DMR overlapping with ICR is methylated; if unmethylated, *H19* is expressed. Methylation status of “secondary” IGF2-DMRs in humans varies in different tissues and seems to be involved in regulation of tissue-specific expression of *IGF2* [105,106,107].

Increased *IGF2* expression was detected in inactive demyelinated lesions when compared to normal appearing white matter, but significantly reduced in remyelinating lesions in comparison to inactive demyelinated lesions in post mortem tissues of MS patients, suggesting that *IGF2* among other genes in inactive demyelinated lesions could initiate and/or support remyelination [108]. IGF-2 was shown to serve as a factor potentiating the growth and differentiation of oligodendrocyte progenitor cells in vitro [109] and as a mediator contributing to the effects of glatiramer acetate-reactive Th2 cells on oligodendrocyte progenitor cells in vitro, and perhaps in vivo within the human CNS [110].

### 4.3. MS-Associatеd Individually Imprinted Genes

*DNMT1* gene: *DNMT1* imprinted gene, associated with MS, is located on the chromosome 19 (19p13.2) and encodes DNA Methyltransferase 1. This enzyme is involved in selective methylation of hemi-methylated DNA; it regulates tissue-specific methylation and is also essential for maintenance of progenitor cells in an undifferentiated state in somatic tissues [111]. The DMR is located at the promoter of *DNMT1* gene and is specifically methylated on the maternal allele in human placenta [112]. *DNMT1* expression is significantly downregulated in PBMC of MS patients compared to healthy controls [113].

*PLAGL1* gene: The pleomorphic adenoma gene-like 1 (PLAGL1) is mapped to chromosome 6 (locus 6q24) and is expressed from the paternal allele in both adult and fetal human tissues [114]. *PLAGL1* encodes C2H2 zinc finger protein, acting as a transcription factor or operating as cofactor of other proteins and nuclear receptors, which regulates the production of p21 protein, inhibiting the progression of the cell cycle (reviewed in [115]). In addition, *PLAGL1* region also encodes a paternally expressed ncRNA, *HYMAI* (Hydatidiform Mole Associated And Imprinted), which is transcribed from the first intron of *PLAGL1* gene. Imprinted expression of these genes requires maternal DNA methylation at the PLAGL1-DMR, that induces the correct chromatin profile [116]. Whole transcriptome analysis reveals increased expression of *PLAGL1* in blood leukocytes of patients with RRMS (in relapse) and with SPMS when compared to the control group [117]. Interestingly, along with p53 PLAGL1 was shown to regulate hormone secretion and metabolism in adipose tissue [118,119]. At the same time, the expression of *PLAGL1* in white adipose tissue was shown to be regulated by androgens in rats [120]. Genetic and epigenetic alterations of this gene have been associated with transient neonatal diabetes mellitus (TNDM), Beckwith–Wiedemann syndrome (BWS), and cancer [115].

The *ZFAT* gene is located on chromosome 8 (locus 8q24.22) and encodes a zinc finger and AT-hook domain containing protein that functions as a suppressor of cell growth. The imprinting *ZFAT* locus also harbors the *ZFAT-AS1* gene—a non-coding antisense RNA overlapping *ZFAT*; both these genes are expressed from the paternal chromosome [121]. It should be mentioned that another study re-established the consistent paternal expression of ZFAT-AS1 in human placenta; the monoallelic expression of the *ZFAT* gene was also revealed, but random activity of either of the parental alleles [122]. It was in one of the genetic studies that the polymorphic variant rs733254 in *ZFAT* gene was associated with RRMS in women, but not in men, in an Arabian Gulf population (odds ratio 2.38 and 95% confidence interval 1.45–3.91; *p* = 0.0014) [123]. Genome-wide association study demonstrated the association of this variant with INF-β therapy response in MS patients [124]. Further studies in mouse revealed that *Zfat* is critical for thymocyte development and T-cells homeostasis in the periphery and that *Zfat* is crucial for the proper expression in peripheral T-cells of Il7rα and Il2rα, which are known MS-susceptible genes [125].

The *RB1* (*Retinoblastoma* gene 1) encodes transcriptional corepressor 1 and is located on chromosome 13 (locus 13q14.2). Imprinting of this gene is caused by the retrotransposition of CpG island from a pseudogene *KIAA0649* on chromosome 9 inside intron 2 of *RB1* (Figure 3) [126]. This CpG sequence serves as an alternative *RB1* promoter and is methylated on the maternal and unmethylated on the paternal chromosome 13. Differential methylation of this CpG island skews the abundance of regular *RB1* transcript from the maternal allele via mechanism of transcriptional interference: The transcription complex is supposed to bind to the unmethylated alternative *RB1* promoter on paternal allele and therefore acts as a roadblock for the regular transcript on the same allele resulting in reduced abundance of paternal *RB1* transcripts (see Figure 3) [126].

A recent study analyzing microarray data demonstrated that *RB1* expression was upregulated in PBMC of RRMS, PPMS, and SPMS patients when compared to healthy controls [127]. Following network analyses identified *RB1* as one of several “network-hub” genes that interacts with many differentially expressed in MS genes as well as regulates their network with MS-associated miRNAs [127]. Safari-Alighiarloo et al. identified lower levels of *RB1* transcripts in cerebrospinal fluid of RRMS patients compared to healthy controls while analyzing array data [128].

The Wilms’ tumor 1 (*WT1*) gene is located on chromosome 11 (locus 11p13) and encodes a zinc-finger-containing transcription factor that is important for normal cellular development and survival. This gene was demonstrated to undergo tissue specific imprinting: Its biallelic expression was shown in the human kidney, while in the fetal brain this gene is exclusively expressed from the maternal allele [129]. A *WT1* paternal imprinting was confirmed in human fibroblasts and lymphocytes in some cases [130]. Genomic imprinting at the *WT1* gene involves its alternative coding transcript (AWT1) that shows deregulation in Wilms’ tumors [131]. The study by Lin et al. indicated that *WT1* intronic variants rs10767935 and rs5030244 may play a role in altering the effects of vitamin D on responses to IFN-β on in MS patients (the group included 71.6% women) [132].

## 5. Conclusions

POEs are now receiving recognition not only in the context of classical diseases of GI, but in a wider range of diseases including complex diseases such as MS. Apparently, this research area is now entering a phase of intensive development. Taking these effects into account in MS studies can help to decipher the mechanisms underlying the associations with disease that are already found and, apparently, partly solve the problem of the “hidden” heritability. To this end, the POE mechanisms in MS are needed to be explored and with that, pay special attention to the sexual dimorphism native to the disease.

This review highlights the protein-coding and non-coding genes from *DLK1-DIO3* and *IGF2-H19* imprinted loci, as well as individually imprinted genes *DNMT1*, *PLAGL1*, *ZFAT*, *RB1*, and *WT1* as promising candidates for uncovering the role of POE in MS. Undoubtedly, POEs in MS should span a much larger spectrum of genes that form a complex ‘network’, which we have yet to discover. Establishing the imprinting status of the known MS-associated loci may be one of the attractive directions to discover. Altogether, these findings will make a significant contribution to the current understanding of MS etiopathology and create new perspectives for MS treatment.

## Figures and Tables

**Figure 1 ijms-22-01346-f001:**
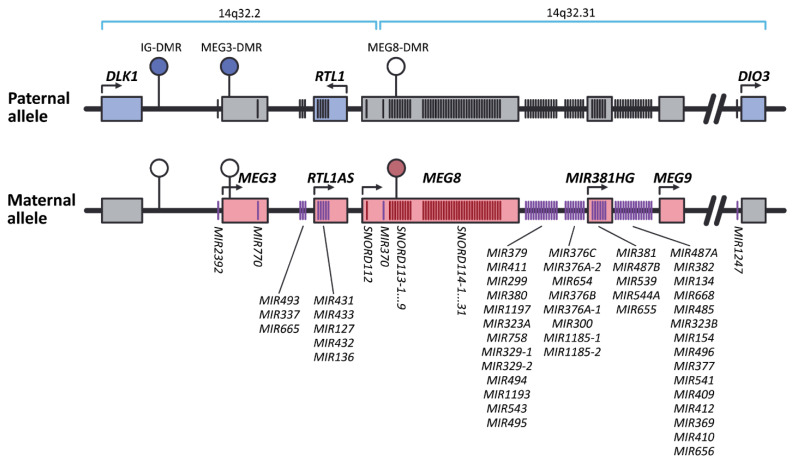
The schematic representation of the locus *DLK1-DIO3* (based on genome assembly GRCh38.p12). The imprinted locus *DLK1-DIO3* contains paternally-expressed protein-coding genes: *DLK1*, *DIO3*, and *RTL1* genes (blue rectangles) and maternally-expressed genes of long noncoding RNAs: *MEG3*, *MEG8*, *MEG9*, and *RTL1AS* (red rectangles). *MEG8* contains a tandemly repeating array of small nucleolar RNAs (snoRNAs) of C/D-box family, namely *SNORD112*, *SNORD113*, and *SNORD114*, consisting of one, nine, and 31 paralogous genes of snoRNAs, respectively. The locus also includes two large clusters of microRNA (miRNA) genes (10 miRNA genes in 14q32.2 and 44 miRNA genes in 14q32.31), expressed from the maternal allele. IG-DMR and MEG3-DMR are methylated on the paternal chromosome, while MEG8-DMR, in contrast, is methylated on the maternal chromosome. Filled ellipses represent methylated DMRs, and open ellipses represent unmethylated DMRs. Gray boxes and black strokes indicate non-expressing genes. Transcriptionally-active genes are marked with colored boxes and strokes; purple and red strokes are miRNA and snoRNA genes, respectively.

**Figure 2 ijms-22-01346-f002:**
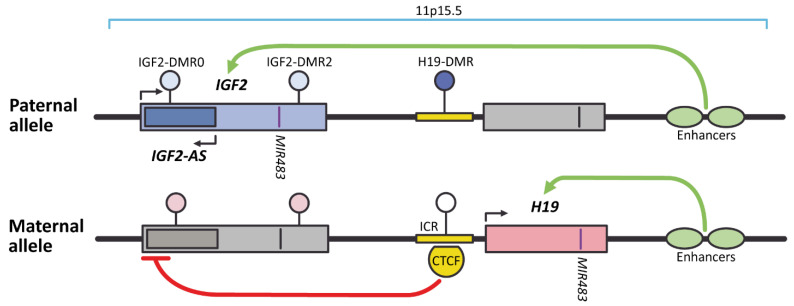
The schematic representation of *IGF2-H19* imprinted locus (based on genome assembly GRCh38.p12). On the paternal allele, methylation of H19-DMR directly blocks the promoter of *H19* gene, but does not prevent 3′ enhancers from activating transcription of *IGF2*, *IGF2-AS*, and *MIR483* genes. On the maternal unmethylated allele, transcriptional repressor CTCF can bind to the ICR overlapping with H19-DMR and block *IGF2* promoter. The enhancer can still efficiently activate the unmethylated *H19* promoter and induce expression of both *H19* and located in it *MIR483*. Dark ellipse represents fully methylated “primary” H19-DMR, open ellipse represent unmethylated “primary” DMR; light colored ellipses are for “secondary” somatic IGF2-DMR0 and IGF2-DMR2, differentially methylated on paternal and maternal chromosomes only in several tissues. Gray boxes and black strokes indicate non-expressing genes. Transcriptionally-active genes are marked with colored boxes and purple strokes.

**Figure 3 ijms-22-01346-f003:**
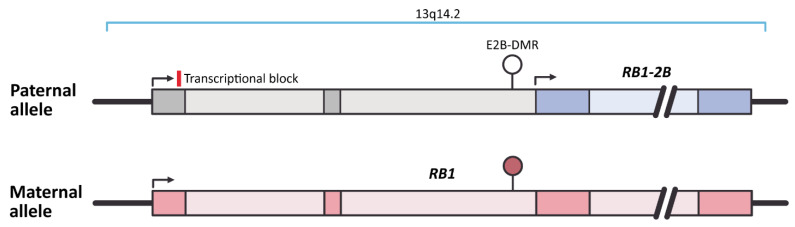
The schematic representation of *RB1* imprinting (based on genome assembly GRCh38.p12). Unmethylated E2B-DMR on the paternal allele activates expression of *RB1-2B*, which acts as a roadblock for the regular *RB1* transcript on the same allele, while methylation of E2B-DMR on maternal allele allows normal expression of *RB1*. Filled ellipse represents methylated E2B-DMR, and open ellipses represent unmethylated DMRs. Dark and light boxes indicate exons and introns of *RB1* gene, respectively.

**Table 1 ijms-22-01346-t001:** Imprinted genes associated with multiple sclerosis (MS).

Gene	Gene Product	Location	Expressed Allele
*DLK1*	Delta Like Non-Canonical Notch Ligand 1	14q32	Paternal
*DNMT1*	DNA Methyltransferase 1	19p13.2 *AS*	Paternal
*IGF2*	Insulin Like Growth Factor 2	11p15.5 *AS*	Paternal
*MEG3*	Long noncoding RNA MEG3 (Maternally Expressed Gene 3)	14q32	Maternal
*PLAGL1*	Pleomorphic Adenoma Gene 1 Protein	6q24.2 *AS*	Paternal
*RB1*	Retinoblastoma Transcriptional Corepressor 1	13q14.2	Maternal
*WT1*	Transcription factor WT1 (Wilms’ Tumor 1)	11p13 *AS*	Paternal
*ZFAT*	Zinc Finger and AT-hook Domain Containing	8q24.22 *AS*	Paternal

**Table 2 ijms-22-01346-t002:** MiRNAs from *DLK1-DIO3* locus associated with MS.

MiRNA	MS Course	Number of Subjects (FEMALE/Male)	Immunomodulatory Treatment (Number of Patients)	Association with MS	Biological Material	Method of Detection	Reference
miR-432-5p	Not specified	14 (not specified)	Not specified (14)	Upregulated in grey matter compared to white matter in MS patients	Brain tissue	NanoString technology	[74]
miR-485	Relapsing-remitting MS (RRMS) (remission)	38/12	Treatment-naive (50)	Downregulated in RRMS patients compared to healthy controls	CD4+ T cells	RT-PCR	[75]
miR-494	RRMS	13/3	Interferon-β (16)	Downregulated in RRMS patients compared to healthy controls	T-cells	Array, RT-PCR	[78]
miR-127-3p, miR-134, miR-136 *, miR-154, miR-299-3p, miR-323-3p, miR-323-5p, miR-323b-3p, miR-329, miR-337-3p, miR-337-5p, miR-369-3p, miR-369-5p, miR-370, miR-370-3p, miR-376a, miR-376b, miR-376c, miR-377, miR-379, miR-381, miR-382, miR-409-3p, miR-409-5p, miR-410, miR-411, miR-431, miR-432, miR-432-5p, miR-433, miR-485-3p, miR-485-5p, miR-487a, miR-487b, miR-493, miR-495, miR-496, miR-539, miR-541, miR-541 *, miR-543, miR-544, miR-654-3p, miR-654-5p, miR-655, miR-656, miR-665, miR-668, miR-758, miR-889, miR-1185, miR-1197	RRMS (remission and relapse)	28/24	Treatment-naive (52)	Upregulated in RRMS patients compared to healthy controls	PBMC	High throughput sequencing, RT-PCR	[44]
miR-381-3p	RRMS	58/20	Fingolimod (78)	Upregulated in fingolimod responders compared with healthy controls	Peripheral blood	RT-PCR	[83]
miR-655-3p				Downregulated in both responders and non-responders compared with healthy controls			
miR-411 *	RRMS	13/4	Natalizumab (17)	Downregulated in RRMS patients compared to RRMS patients upon natalizumab treatment	Peripheral blood	Array	[84]
miR-127-3p, miR-370-3p, miR-409-3p, miR-432-5p	Primary progressive MS (PPMS), secondary progressive MS (SPMS)	5/1 (PPMS), 10/6 (SPMS)	Dimethyl fumarate (3) Fingolimod (2) Interferon beta (3) Natalizumab (2) Glatiramer acetate (1)	Upregulated in SPMS/PPMS patients compared to healthy controls	Serum	High throughput sequencing	[77]
miR-376c-3p	PPMS	18/13	No immunomodulatory therapy within 6 months prior to blood sampling	Upregulated in PPMS patients compared to healthy controls	Serum	PCR-array, RT-PCR	[85]
miR-432-5p, miR-433-3p, miR-485-3p	RRMS, PPMS, SPMS	10/4 (RRMS), 5/1 (PPMS), 10/6 (SPMS)	Dimethyl fumarate (6) Fingolimod (3) Interferon beta (3) Natalizumab (3) Glatiramer acetate (1)	Upregulated in SPMS/PPMS patients compared to RRMS	Serum	High throughput sequencing	[77]
miR-337-3p	RRMS, SPMS	60/32 (RRMS), 24/6 (SPMS)	No disease-modifying treatment at the time of blood sampling	Negatively correlates with the EDSS in RRMS and SPMS patients	Serum	RT-PCR	[79]
miR-300	RRMS, SPMS	28/11 (RRMS), 24/6 (SPMS)	RRMS: Fingolimod (7), Interferon-β (18), Azathioprine (12), Methotrexate (2). SPMS: Fingolimod (11), Interferon-β (8), Azathioprine (15), Methotrexate (1)	Downregulated in MS patients compared to healthy controls	Serum	RT-PCR	[86]

## Data Availability

Not applicable.

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
