# Peer review of "Imprinted Genes and Multiple Sclerosis: What Do We Know?"

_ijms, 2021, doi:10.3390/ijms22031346_

Round 1

Reviewer 1 Report

In this review the authors collected the knowledge about Imprinted genes and multiple sclerosis. This is a hot topic and even there are a lot of paper published in this field, too much is yet unknown. Nevertheless, in my opinion the manuscript needs a minor revision to better highlight the works and topics. Since the authors didn't follow the PRISMA guidelines maybe because is not a systematic review, they should at least explain the procedure that they applied to select the genes to investigate. Is clear that the first screening was make by geneimprint database website, but starting from 107 genes then is lack the information or is not clear how they arrive to select 8 and then 3 genes that were well deep investigated.

About table 2, is a bit confused and need to be reorganize.First of all in the column  "Association with MS" it would be better decide which information insert and write the same for all papers select (for i.e., stage of the disease and drug treatment).

Finally, the "conclusions section" in my opinion need more discussion try to highlight what is the importance and how to use the informations collected in this review from the future scientists that will study in this field.

Author Response

- Reviewer 1. In this review the authors collected the knowledge about Imprinted genes and multiple sclerosis. This is a hot topic and even there are a lot of paper published in this field, too much is yet unknown. Nevertheless, in my opinion the manuscript needs a minor revision to better highlight the works and topics.

Dear Reviewer,

Thank you for the careful and thoughtful analysis of our manuscript and for the remarks of essential significance. We are sincerely grateful for all your advices. Please, find below the responses to each of your comments, step by step.  All changes in the revised manuscript made in accordance with your recommendations are marked.

- Since the authors didn't follow the PRISMA guidelines maybe because is not a systematic review, they should at least explain the procedure that they applied to select the genes to investigate. Is clear that the first screening was make by geneimprint database website, but starting from 107 genes then is lack the information or is not clear how they arrive to select 8 and then 3 genes that were well deep investigated.

Following your remark, we now modified the description of the procedure we applied to select the genes for further consideration (first abstract of “Imprinted genes and MS” chapter, p.3). Now it sounds as:

“We performed in the PubMed database [https://pubmed.ncbi.nlm.nih.gov/] a search of studies (regardless of the year of their publication), which are indexed by MeSH terms "Multiple sclerosis" and the name of each of these 107 imprinted genes. For further consideration, we selected those genes that were mentioned in publications fulfilling the following criteria: 1) the publication is an original article; 2) the publication contains information on the association of the MeSH-gene with MS and/or with its animal models; 3) biological materials from humans and/or animal model of MS were used to confirm this association. As a result, eight genes with known imprinted status were selected, among which 6 genes, DLK1, DNMT1, IGF2, MEG3, PLAGL1 and ZFAT are paternally expressed (75%), and 2 genes, RB1 and WT1 - maternally expressed (Table 1). Here we will consider the association with MS of all these genes. Based on the genomic organization they could be divided into those that are components of imprinted loci or single-imprinted. Of all these genes only 3 are located in imprinted loci: DLK1 and MEG3 are clustered in DLK1-DIO3 locus, and IGF2 - in IGF2-H19 locus. Due to the common mechanisms of regulation in an imprinted locus, we will also highlight the currently known data on the involvement of other components of DLK1-DIO3 and IGF2-H19 loci in MS, since it may promote the interest for understanding the role of GI in MS”.

- About table 2, is a bit confused and need to be reorganize. First of all in the column "Association with MS" it would be better decide which information insert and write the same for all papers select (for i.e., stage of the disease and drug treatment).

Thank you for this recommendation that improved the informativity and presentability of the Table 2. We now reorganized its structure and added 4 new columns, describing MS course, Number of subjects with female/male detailing, Drug treatment with number of patients and Method of detection (pp. 7-9).

- Finally, the "conclusions section" in my opinion need more discussion try to highlight what is the importance and how to use the informations collected in this review from the future scientists that will study in this field.

We provided feedback on your recommendations in the “Conclusion” section (p.14) and added the following sentences:

“Taking these effects into account in MS studies can help to decipher the mechanisms underlying the associations with disease that are already found and, apparently, partly solve the problem of the 'hidden' heritability”.

And:

“This review highlights the protein-coding and non-coding genes from DLK1-DIO3 and IGF2-H19 imprinted loci, as well as individually imprinted genes DNMT1, PLAGL1, ZFAT, RB1, and WT1 as promising candidates for uncovering the role of POE in MS. Undoubtedly, POEs in MS should span a much larger spectrum of genes that form a complex 'network', which we have yet to discover. Establishing the imprinting status of the known MS-associated loci may be one of the attractive directions to discover. Altogether, these findings will make a significant contribution to the current understanding of MS etiopathology and create new perspectives for MS treatment”.

Reviewer 2 Report

This is an extensive review concerning the potential role of genomic imprinting in Multiple Sclerosis etiology.  The hypothesis of epigenetic mechanisms which modulates MS clinical presentation and progression is not new, but the review focused  on the parent of origin effect and specifically on genomic imprinting that is recently recognized as important pathogenetic mechanism in several complex diseases. The paper is well written and it’s worth reporting. The authors reviewed a huge amount of data and presented a photograph of what we know today about parent of origin effect in MS. The review can be useful for other studies to improve our knowledge of MS molecular mechanisms.  

Author Response

Dear Reviewer, thank you for the appreciation of our work!

Reviewer 3 Report

It is an interesting manuscript that summarized the current views on the potential role of imprinted genes in the pathophysiology of multiple sclerosis.

The referee has some minor suggestions for the authors:

  • Since it is a review, it would be helpful to add a paragraph regarding the visited databases, the employed keywords, the selection criteria, and the review flowchart (number of studies screened, number of study after removing duplicate, and number of studies analyzed).

  • The manuscript would benefit from some editing.

  • It is important to add references for some sentences that lack citations (e.g., l.52-55).

  • Section 2, l.52-55: it is preferable to paraphrase this sentence since it resembles the paragraph from Rampersaud E et al. Curr Diabetes Rev. 2008; 4:329-39.